# NORMALIZED MATCHING TRANSFORMER

## ABSTRACT

We introduce the **Normalized Matching Transformer (NMT)**, a deep learning approach for efficient and accurate sparse keypoint matching between image pairs. NMT consists of a strong visual backbone, geometric feature refinement via SplineCNN, followed by a normalized transformer for computing matching features. Central to NMT is our *hyperspherical normalization strategy*: we enforce unit-norm embeddings at every transformer layer and train with a combined contrastive InfoNCE and hyperspherical uniformity loss to yield more discriminative keypoint representations. This novel architecture/loss combination encourages close alignment of matching image features and large distance between non-matching ones not only at the output level, but for each layer. Despite its architectural simplicity, NMT sets a new state-of-the-art performance on PascalVOC and SPair-71k, outperforming BBGM Rolínek et al. (2020), ASAR Ren et al. (2022), COMMON Lin et al. (2023) and GMTR Guo et al. (2024) by 5.1% and 2.2%, respectively, while converging in at least $\geq 1.7\times$ fewer epochs compared to other state of the art baselines. These results underscore the power of combining pervasive normalization with hyperspherical learning for geometric matching tasks.

Code will be made publicly available upon acceptance.

## 1 INTRODUCTION

Traditional graph-matching pipelines Rolínek et al. (2020); Torresani et al. (2012) rely on neural network backbones for computing discriminative features combined with combinatorial solvers to establish keypoint correspondences to address the feature matching problem. While effective, these approaches are often complex, requiring the combination of a neural network based feature computation stage with an intricate combinatorial stage for computing keypoint correspondences. Integrating the combinatorial stage into a neural network pipeline brings its own challenges, including non-differentiability and most often combinatorial solvers running on CPUs. Recent methods like GMTR Guo et al. (2024), ASAR Ren et al. (2022) and COMMON Lin et al. (2023) proposed pure deep-learning approaches with a simpler Sinkhorn-based decoding. They have sought to enhance performance and robustness through transformer-based architectures, better losses and/or specialized regularization strategies. These newer approaches, while foregoing a combinatorial stage, still outperform hybrid approaches, attesting to the strength of deep learning even in the setting of keypoint matching that has a strong combinatorial aspect to it.

While pure deep learning methods have already reached very high results on keypoint matching datasets, we show that there is still room for improvement. First, better backbones boost performance. We replace the commonly used VGG Simonyan & Zisserman (2014) backbone for a current swin-transformer Liu et al. (2021). Second, we process features at keypoints with a SplineCNN GNN Fey et al. (2018), which adds helpful inductive biases incorporating the geometry of the keypoints to match. Third, we use transformers to mix information between and across images. We show that vanilla transformers can be outperformed by additional normalization techniques used in normalized transformers Loshchilov et al. (2024). We argue that the employed normalization techniques are well-suited for our normalized feature representation: Instead of only normalizing at the end before doing cosine similarity and computing the losses, we normalize throughout the normalized transformer, which helps in faster training and better overall performance. Our pipeline is trained using a contrastive Oord et al. (2018) and hyperspherical Mettes et al. (2019) loss together with data augmentation.

Figure 1: Normalized matching transformer inference. A pair of images is passed each through a swin-transformer visual backbone. Features at keypoints are extracted and given through a SplineCNN for further feature refinement. Two normalized transformer decoders interleave self-attention between keypoint features from the same image with cross-attention that mixes information across images. Finally, cosine similarities are computed and given as affinities to a logspace Sinkhorn routine from which a matching is decoded.

Our work shows that the performance for keypoint matching, one of the classical and very well-studied problems in computer vision, has not yet saturated and can be enhanced by leveraging current deep learning methods. In particular, we argue that our contribution of hyperspherical architecture and lossses enhances feature quality and improves training speed. Also no combinatorial subroutines are necessary given the capabilities of our neural network pipeline, simplifying our overall approach.

**Contribution.** In detail, our contributions are as follows:

**Architecture:** We propose a simple and efficient pure ML-based architecture combining an image processing backbone using a swin-transformer Liu et al. (2021), followed by a keypoint feature processing stage consisting of SplineCNN Fey et al. (2018), a graph neural network that exploits the geometrical structure of keypoints.

**Decoding:** The feature computation stage is followed by a two-stream transformer decoder, each stream processing keypoints from one image. Each decoder employs normalized transformer layers Loshchilov et al. (2024) with unit-norm parameter normalization and decoupled attention and MLP residual pathways to stabilize gradient flow throughout the network. Raw cosine-similarity affinities are computed via the normalized self-attention outputs. Only during inference, a differentiable Sinkhorn algorithm Cuturi (2013) converts these affinities into a soft, doubly-stochastic matching matrix. This design removes the need for combinatorial solvers and yields a streamlined, end-to-end matching pipeline.

**Loss:** Our method incorporates improved loss formulations, including InfoNCE Oord et al. (2018) and hyperspherical loss Mettes et al. (2019). They improve feature embedding quality and ensure robust training, enabling the model to learn more discriminative representations.

**Experimental:** Extensive experiments demonstrate state-of-the-art performance on PascalVOC and SPair-71k datasets, with significant improvements over the state of the art methods BBGM Rolínek et al. (2020), ASAR Ren et al. (2022), COMMON Lin et al. (2023) and GMTR Guo et al. (2024) exceeding their performance by 5.1% on PascalVOC and 2.2% on SPair-71k. We also need at least $1.7x$ fewer epochs until training convergence than the baselines.

## 2 RELATED WORK

Related work on keypoint matching involves (i) *combinatorial* aspects for establishing a one-to-one correspondence between sets of keypoints, (ii) *hybrid* approaches that combine mainly neural networks for computing keypoint features with combinatorial routines to get correspondences and (iii) *pure deep learning* based methods that forego any combinatorial subroutines.

On the application side we distinguish between (i) *sparse* keypoint matching, which we study here, for computing correspondences between few select keypoints of distinct objects of the same class in different environments and (ii) *dense* keypoint matching for estimating homographies between many keypoints belonging to the same object in the same scene but e.g. viewed from different viewpoints.

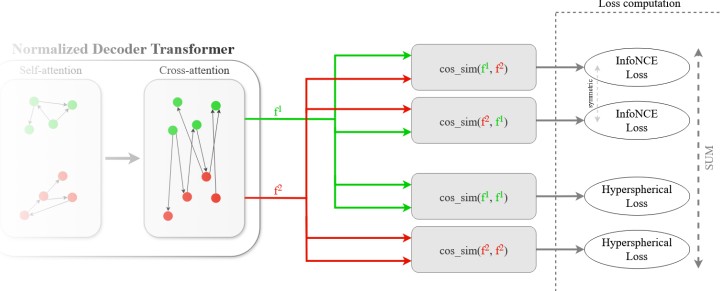

Figure 2: Normalized matching transformer losses. Losses are applied on the features that are computed by the normalized transformer decoder. InfoNCE losses are computed on cosine similarities of features coming from a single keypoint in one image and all keypoint features from the other one and align matching correspondences. For symmetry we apply the InfoNCE loss in both directions. For distributing features of different keypoints in the same image we use a hyperspherical loss on the keypoint features coming from each image separately.

**Combinatorial Aspects & Assignment Problems.** In the combinatorial literature the task of finding one-to-one correspondences between sets of points is called the assignment problem. When the cost function consists of only terms that measure how well two points match onto each other, we obtain the linear assignment problem. This problem is polynomially solvable and fast solvers for practical problems exist Ahuja et al. (1993). The Sinkhorn algorithm Cuturi (2013), an approximation to the linear assignment problem, is especially popular in machine learning, since it is easy to implement, can be differentiated through and runs on GPUs.

When additionally pairwise terms are used that measure how well pairs of points match to each other we obtain the quadratic assignment problem, also known as graph matching in the computer vision literature. In the keypoint matching scenario the quadratic assignment problem allows to incorporate geometric information, e.g. penalizing matching keypoints that are nearby in one image to ones that are far away from each other in the other image etc. From a computational standpoint, the quadratic assignment problem is much more involved. Current state of the art solvers Torresani et al. (2012); Zhang & Lee (2019); Swoboda et al. (2017); Hutschenreiter et al. (2021); Haller et al. (2022); Kahl et al. (2024) are complex and, while relatively fast, still present a computational bottleneck. From a theoretical viewpoint the quadratic assignment problem is a well known NP-hard problem Lawler (1963) and notoriously difficult in practice Burkard et al. (1997).

**Hybrid Approaches.** For the keypoint matching problem the traditional approach is to first extract discriminative features for each keypoint (resp. for pairs of keypoints), use those to compute costs for matching keypoints and finally to compute correspondences using the linear or quadratic assignment problem. Some approaches use ad-hoc heuristics for decoding correspondences, e.g. via reformulation to constrained vertex classification Wang et al. (2021) or the quadratic assignment problem Torresani et al. (2012); Rolínek et al. (2020); Wang et al. (2021).

Pre-neural network approaches with hand-crafted feature descriptors were for quite some time still state of the art Torresani et al. (2012). However, neural network features eventually overtook Zanfir & Sminchisescu (2018). Follow-up work NGM Wang et al. (2021) differentiates through the construction of a quadratic assignment problem and decodes the matching by converting to a constrained vertex classification problem. The hybrid approach Rolínek et al. (2020) combined a state of the art neural network pipeline with a quadratic assignment solver and used a special backpropagation technique Vlastelica et al. (2019) to learn in tandem with the non-differentiable combinatorial solver.

**Pure Deep-Learning.** When not using combinatorial routines it is even more important to obtain discriminative features. One of the first pure neural network methods Zanfir & Sminchisescu (2018) relaxed a graph matching solver to be differentiable and used feature hierarchies. PCA Wang et al. (2019) differentiates end-to-end and learns linear and quadratic affinity costs. QCDGM Gao et al. (2021) proposes a differentiable quadratic constrained-optimization compatible with a deep learn-

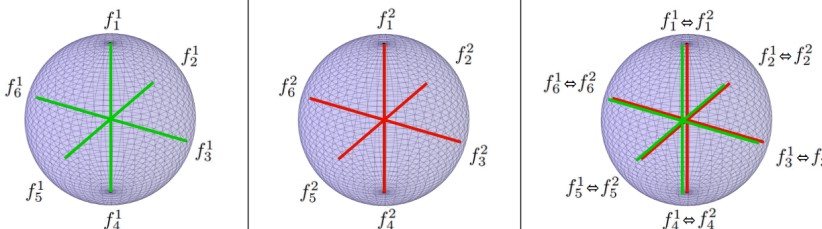

Figure 3: Geometric illustration of hyperspherical and infoNCE losses. The hyperspherical losses (two left spheres) from equation 3 distributes different keypoint features $f_j^i$ for different keypoints $j \in [m]$ and each image $i \in [2]$ across the hypersphere and is applied to each image separately. The InfoNCE (right side) loss from equation 1 aligns features $f_j^1 \Leftrightarrow f_j^2$ from matching keypoints (assuming the matching is identity here) from different images.

ing optimizer and a balancing term in the loss function GLMNet Jiang et al. (2019) utilizes a GNN and alleviates oversmoothing by utilizing an anistropic Laplacian "sharpening" operation. CIE Yu et al. (2019) employs attention and improves upon plain attention by enforcing channel independence and sparsity in the in ensuing matching decoding step. DGMC Fey et al. (2020) uses the graph neural network Fey et al. (2018) and an ad-hoc message passing routine for obtaining correspondences. COMMON Lin et al. (2023) likewise uses SplineCNN and trains using contrastive losses. ASAR Ren et al. (2022) improves performance by using adversarial training and an advanced regularization technique. GMTR Guo et al. (2024) uses a transformer architecture with self- and cross-attention to exchange information between keypoints in the same and different images.

Our approach differs from previous work in jointly combining a strong backbone, transformer based architecture with contrastive and hyperspherical representation learning and hyperspherical normalization throughout the transformer. These were not or not jointly used before. We show that this combination achieves better performance while needing a shorter training.

In the closely related area of dense keypoint matching, transformer based architectures Sarlin et al. (2020); Lindenberger et al. (2023); Sun et al. (2021) have become state of the art as well.

## 3 METHOD

Our method consists of five building blocks:

**Visual feature extraction:** Visual features are extracted from the images using a pre-trained swin-transformer Liu et al. (2021) backbone.

**GNN:** Keypoints are treated as nodes to construct an undirected graph from the visual features. A SplineCNN Fey et al. (2018) is employed to refine the node features by aggregating local spatial information. This approach was pioneered in Fey et al. (2020).

**Normalized transformer:** The normalized Transformer (nGPT) Loshchilov et al. (2024) architecture is used with self-attention layers for exchanging information between keypoints of the same image and cross-attention layers for exchanging information between keypoints from different layers.

**Sinkhorn matching:** Using the refined features we compute cosine similarities between pairs of keypoints from each image. This similarity score matrix is passed through a logspace-sinkhorn algorithm for ensuring a double stochastic matrix. Each index of the maximum in each row indicates the best possible match between the source (row) and target (column) node.

**InfoNCE and hyperspherical losses:** We use InfoNCE Oord et al. (2018) and hyperspherical Melekhov et al. (2019) loss functions for shaping better feature representations. InfoNCE aligns features for the corresponding keypoints in different images and penalizes alignment of non-corresponding features. The hyperspherical loss distributes keypoint features from the same image uniformly on the hypersphere, ensuring more distinctive fea-

tures. For better results we apply the hzperspherical loss after each normalized transformer layer.

An illustration of our approach is provided in Figure 1 for the inference and 2 and 3 for losses during training.

**Visual Feature Extraction.** We are given two images $I^1, I^2$ alongside coordinates $k_1^1, \ldots, k_m^1$ and $k_1^2, \ldots, k_m^2$ specifying the $m$ keypoint positions in each image We pass both images through a swin-transformer Liu et al. (2021) and obtain down-sampled features. The spatial features corresponding to the specified keypoints are interpolated from these downsampled features using a bilinear sampling technique. For each keypoint we extract features from the last and second last layer of the backbone and concatenate them. Following BBGM Rolínek et al. (2020) we additionally mean-pool all features from the backbone to get a global feature for each image that helps to class-condition the matching process.

We use the swin-large version as backbone.

**GNN.** To incorporate the spatial structure of the objects as indicated by the geometry of the keypoints, we use a SplineCNN Fey et al. (2018) as suggested in Fey et al. (2020). We construct a graph with nodes being keypoints and edges coming from a Delaunay triangulation. Two rounds of graph convolution are performed to refine feature representations for each image separately. Unlike general-purpose GNNs that focus on aggregating features across nodes without explicit consideration of their spatial layout, SplineCNN utilizes spline-based kernels that adapt to the graph's Euclidean geometry, allowing for a better representation of spatial interactions.

The GNN operates on a message-passing paradigm, where node features are iteratively updated by aggregating information from neighboring nodes at each layer. Trainable B-Spline kernels anisotropically convolve features from other nodes.

Our implementation employs two spline convolution layers, each with a kernel size of 5 and utilizes max aggregation. We use ReLU as non-linearity. We use Euclidean coordinates for the geometric input to the trainable B-splines.

**Normalized Transformer.** The normalized transformer Loshchilov et al. (2024), a variant of the original transformer architecture Vaswani (2017), uses hyperspherical normalization and projects intermediate and final representations back to unit norm. This aligns well with the graph matching setting where we want to measure potential correspondences by cosine similarity of their (implicitly normalized) features. Experiments in NLP have demonstrated that the normalized transformer architecture can converge faster, be numerically more stable and can reach better solutions.

In particular, the normalized transformer uses normalization after attention, the MLP block and the residual connections. Let $f = f_1, \ldots, f_m$ be a feature sequence coming from all the keypoints in an image. For cross-attention let $f_{other}$ be the keypoint feature sequence from the other image. Then normalized self-attention, cross-attention and MLP layers can be written as:

---
`Norm.Self-Attn`$(f)$:

$f_A \leftarrow \text{Norm}(\text{Self-Attn}(f))$
$f \leftarrow \text{Norm}(f + \alpha_A \cdot (f_A - f))$

---

`Norm.Cross-Attn`$(f, f_o)$:

$f_A \leftarrow \text{Norm}(\text{Cross-Attn}(f, f_o))$
$f \leftarrow \text{Norm}(f + \alpha_C \cdot (f_A - f))$

---

`Norm.MLP`$(f)$:

$f_M \leftarrow \text{Norm}(\text{MLP}(f))$
$f \leftarrow \text{Norm}(f + \alpha_M \cdot (f_M - f))$

---

Element-wise step sizes $\alpha_A, \alpha_C, \alpha_M$ in residual connections are learned positive vectors.

A transformer layer in our matching method consists of first doing normalized self-attention separately for keypoint features of each image, then doing two normalized cross-attention passes where one keypoint feature sequence attends to the other respectively and finally passing through a normalized MLP block. Additionally, after cross-attention we element-wise modulate keypoint features with the global feature token and normalize afterwards to unit norm again.

We use 4 such two-stream normalized transformer decoder layers using 12 heads with a feature dimension of 648. We use SiLU activations.

**Matching.** To establish correspondences between keypoints, we first compute cosine similarities between each pair of features coming from different images. During Inference this square affinity matrix is given to the Sinkhorn algorithm, which outputs a double stochastic matrix. To decode final correspondences, we use the computed double stochastic matrix and go through each row $i$ and pick the column $j$ with maximum entry, meaning that keypoint $i$ in the first image is matched to keypoint $j$ in the second one.

Our full matching pipeline is detailed in the Normalized Matching Transformer algorithm.

---

```
Normalized Matching Transformer
```
---

**Input:** Input images $I^1, I^2$,
keypoints $k_1^1, \ldots, k_m^1, k_1^2, \ldots, k_m^2$
**Output:** Matching $\pi : [m] \to [m]$
```
// Swin-transformer backbone
```
$g_1^i, \ldots, g_n^i = \text{Backbone}(I^i)$ $\qquad\qquad \forall i \in [2]$
```
// Global feature token
```
$f_{global}^i = \text{Avg-Pool}(f_1^i, \ldots, f_n^i)$ $\qquad\qquad \forall i \in [2]$
```
// Interpolate features at keypoint
```
$f_j^i = \text{Interp}(g^i, k_j),$ $\qquad\qquad \forall i \in [2], j \in [m]$
```
// SplineCNN GNN
```
$f^i = f_1^i, \ldots, f_m^i = \text{GNN}(f_1^i, \ldots, f_m^i)$ $\qquad\qquad \forall i \in [2]$
```
// Normalized Transformer Decoder
```
**for** $iter = 1, \ldots, L$ **do**
$\quad f^i = \texttt{Norm.Self-Attn}(f^i, f_{global}^i)$
$\quad f^1 = \texttt{Norm.Cross-Attn}(f^1, f^2)$
$\quad f^2 = \texttt{Norm.Cross-Attn}(f^2, f^1)$
$\quad f_j^i = \text{Norm}(f_j^i \cdot f_{global}^i)$ $\qquad\qquad i \in [2], j \in [m]$
$\quad f^i, f_{global}^i = \texttt{Norm.MLP}(f^i, f_{global}^i)$ $\qquad\qquad \forall i \in [2]$
```
// Sinkhorn matching
```
$C_{ij} = \cos\text{sim}(f_i^1, f_j^2)$ $\qquad\qquad \forall i, j \in [m]$
$A = \text{Sinkhorn}(C)$
$\pi(i) = \arg\max_{j \in [n]}\{A_{ij}\}$ $\qquad\qquad \forall i \in [m]$

---

**Training Losses.** We use the contrastive InfoNCE loss introduced in Oord et al. (2018) for better feature representations. Corresponding points are treated as positive pairs, while non-corresponding matches from the other image are treated as negative ones. This leads to matching keypoints having aligned representations with large cosine similarity while non-matching ones having low cosine similarity.

In particular, let $f_i^1$ and $f_j^2$ be two matching keypoints features coming out of the transformer decoder. Let $f_i^1$ and $f_l^2, l \in [m]\backslash\{j\}$ be the keypoint features in image 2 that do not match to $f_i^1$. Then the InfoNCE loss is

$$\mathcal{L}_{\text{InfoNCE}} = -\log \frac{\exp(\cos\text{sim}(f_i^1, f_j^2)/\tau)}{\sum_{l \in [m]\backslash\{j\}} \exp(\cos\text{sim}(f_i^1, f_l^2)/\tau)}, \qquad (1)$$

Here $\tau > 0$ is a learnable parameter. The overall InfoNCE loss is then the summation of the InfoNCE losses over all keypoints features. To symmetrize, we also take the matches the other way around.

To further encourage separation between keypoint features coming from the same image and to promote a more uniform distribution of features on the hypersphere we use a hyperspherical loss Mettes et al. (2019). Let

$$C = \big(\cos \operatorname{sim}(f_i^1, f_j^2)\big)_{i,j \in [m]} \in \mathbb{R}^{m \times m} . \tag{2}$$

be the matrix of all pairwise cosine similarities. Then the hyperspherical loss is

$$\mathcal{L}_{\mathbf{HS}} = \sum_{j=1}^{n} \max_{j \neq i} C_{ij} . \tag{3}$$

This loss penalizes whenever two different keypoints are aligned. The $\max_{j \neq i}$ ensures that this penalization is not carried out for the same keypoint.

We also incorporate the hyperspherical loss as an auxiliary layer loss on every matching transformer layer. In our approach, the loss is weighted using a parameter $p = 0.3$ that increases linearly with the layer depth . Then the loss is

$$\mathcal{L}_{\mathbf{HS}}^{\mathbf{layer}} = \sum_{k=1}^{L} k\mathrm{p} \cdot \mathcal{L}_{\mathbf{HS}}^{(k)} , \tag{4}$$

where $p$ is the weighted hyperparameter and $\mathcal{L}_{\mathbf{HS}}^{(k)}$ is the layer wise hyperspherical loss. This progressive weighting ensures that deeper layers, which contribute more critically to the final feature representations, receive a stronger regularization. We then average over the two decoders to obtain a single layer-wise loss and add that to the overall hyperspherical loss computed from the final decoder outputs. Consequently, this strategy encourages a more uniform distribution of keypoint features on the hypersphere across all layers, ultimately leading to enhanced feature distinctiveness and improved matching performance.

Our overall loss sums up the InfoNCE and hyperspherical loss without any weighting.

An illustration of the loss construction is given in Figures 2 and 3.

## 4 EXPERIMENTS

Table 1: Default hyperparameters for swin transformer, splineCNN, normalized transformer, and training settings.

(a) Swin-Large

| Parameter | Value |
|---|---|
| image size | 384 |
| patch size | 4 |
| window size | 24 |
| embedding dimension | 128 |

(b) SplineCNN

| Parameter | Value |
|---|---|
| input features | 1024 |
| output features | 648 |
| # layers | 2 |
| kernel size | 5 |

(c) Transformer

| Parameter | Value |
|---|---|
| model dim, $d_{\mathrm{model}}$ | 648 |
| # heads | 12 |
| # decoder layers | 4 |
| MLP hidden mult | 4 |
| Activation | SiLU |
| layer loss param | 0.3 |

(d) Training

| Parameter | Value |
|---|---|
| Batch size (PascalVOC) | 8 |
| Batch size (SPair-71k) | 5 |
| # Epochs | 6 |
| Learning rate | $5 \times 10^{-4}$ |

Table 2: Average accuracy (%) of each object category on Pascal VOC. Best results are **bold** and second best are underlined.

| Method | ✈ | 🚲 | 🐦 | 🚤 | 🍾 | 🚌 | 🚗 | 🐱 | 🪑 | 🐄 | 🪑 | 🐕 | 🐴 | 🏍 | 🚶 | 🪴 | 🐑 | 🛋 | 🚆 | 🖥 | Mean |
|---|---|---|---|---|---|---|---|---|---|---|---|---|---|---|---|---|---|---|---|---|---|
| GMN-PL Patil et al. (2021) | 31.1 | 46.2 | 58.2 | 45.9 | 70.6 | 76.5 | 61.2 | 61.7 | 35.5 | 53.7 | 58.9 | 57.5 | 56.9 | 49.3 | 34.1 | 77.5 | 57.1 | 53.6 | 83.2 | 88.6 | 57.9 |
| PCA Wang et al. (2019) | 40.9 | 55.0 | 65.8 | 47.9 | 76.9 | 77.9 | 63.5 | 67.4 | 33.7 | 66.5 | 63.6 | 61.3 | 58.9 | 62.8 | 44.9 | 77.5 | 67.4 | 57.5 | 86.7 | 90.9 | 63.8 |
| NGM Wang et al. (2021) | 50.8 | 64.5 | 59.5 | 57.6 | 79.4 | 76.9 | 74.4 | 69.9 | 41.5 | 62.3 | 68.5 | 62.2 | 62.4 | 64.7 | 47.8 | 78.7 | 66.0 | 63.3 | 81.4 | 89.6 | 66.1 |
| GLMNet Jiang et al. (2019) | 52.0 | 67.3 | 63.2 | 57.4 | 80.3 | 74.6 | 70.0 | 72.6 | 38.9 | 66.3 | 77.3 | 65.7 | 67.9 | 64.2 | 44.8 | 86.3 | 69.0 | 61.9 | 79.3 | 91.3 | 67.5 |
| CIE Yu et al. (2019) | 51.2 | 69.2 | 70.1 | 55.0 | 82.8 | 72.8 | 69.0 | 74.2 | 39.6 | 68.8 | 71.8 | 70.0 | 71.8 | 66.8 | 44.8 | 85.2 | 69.9 | 65.4 | 85.2 | 92.4 | 68.9 |
| DGMC Fey et al. (2020) | 50.4 | 67.6 | 70.7 | 70.5 | 87.2 | 85.2 | 82.5 | 74.3 | 46.2 | 69.4 | 69.9 | 73.9 | 73.8 | 65.4 | 51.6 | 98.0 | 73.2 | 69.6 | 94.3 | 89.6 | 73.2±0.5 |
| BBGM Rolínek et al. (2020) | 61.5 | 75.0 | 78.1 | 80.0 | 87.4 | 93.0 | 89.1 | 80.2 | 58.1 | 77.6 | 76.5 | 79.3 | 78.6 | 78.8 | 66.7 | 97.4 | 76.4 | 77.5 | 97.7 | 94.4 | 80.1±0.6 |
| GMTR Guo et al. (2024) | 69.0 | 74.2 | 84.1 | 75.9 | 87.7 | 94.2 | 90.9 | 87.8 | 62.7 | 83.5 | 93.9 | 84.0 | 78.7 | 79.6 | 69.2 | **99.3** | 82.5 | 83.0 | **99.1** | 93.3 | 83.6 |
| COMMON Lin et al. (2023) | 65.6 | 75.2 | 80.8 | 79.5 | 89.3 | 92.3 | 90.1 | 81.8 | 61.6 | 80.7 | **95.0** | 82.0 | 81.6 | 79.5 | 66.6 | 98.9 | 78.9 | 80.9 | **99.3** | 93.8 | 82.7 |
| NMT (ours) | **75.8** | **81.9** | **90.9** | **82.4** | **93.5** | **95.4** | **92.7** | **90.7** | **84.6** | **85.2** | 92.9 | **89.3** | **89.4** | **86.9** | **77.2** | 98.5 | **85.8** | **88.0** | 97.6 | **95.5** | **88.7** |

Table 3: Average accuracy (%) of each object category on SPair-71k. Best results are **bold** and second best are underlined.

| Method | ✈ | 🚲 | 🐦 | 🚤 | 🍾 | 🚌 | 🚗 | 🐱 | 🪑 | 🐄 | 🐕 | 🐴 | 🏍 | 🚶 | 🪴 | 🐑 | 🚆 | 🖥 | Mean |
|---|---|---|---|---|---|---|---|---|---|---|---|---|---|---|---|---|---|---|---|
| DGMC Fey et al. (2020) | 54.8 | 44.8 | 80.3 | 70.9 | 65.5 | 90.1 | 78.5 | 66.7 | 66.4 | 73.2 | 66.2 | 66.5 | 65.7 | 59.1 | 98.7 | 68.5 | 84.9 | 98.0 | 72.2 |
| BBGM Rolínek et al. (2020) | 66.9 | 57.7 | 85.8 | 78.5 | 66.9 | 95.4 | 86.1 | 74.6 | 68.3 | 78.9 | 73.0 | 67.5 | 79.3 | 73.0 | 99.1 | 74.8 | 95.0 | 98.6 | 78.9 |
| GMTR Guo et al. (2024) | 75.6 | 67.2 | 92.4 | 76.9 | 69.4 | 94.8 | 89.4 | 77.5 | 72.1 | 86.3 | 77.5 | 72.2 | **86.4** | 79.5 | 99.6 | **84.4** | 96.6 | 99.7 | 83.2 |
| COMMON Lin et al. (2023) | 77.3 | 68.2 | 92.0 | 79.5 | 70.4 | 97.5 | 91.6 | **82.5** | 72.2 | **88.0** | 80.0 | 74.1 | 83.4 | **82.8** | **99.9** | 84.4 | 98.2 | 99.8 | 84.5 |
| NMT (ours) | **79.3** | **72.7** | **94.9** | **84.2** | **74.8** | **98.7** | **94.4** | 82.2 | **81.1** | **88.0** | **85.5** | **81.3** | 82.3 | 79.4 | 100 | 83.2 | **99.2** | **99.9** | **86.7** |

**Training Details.**  Our network is trained using the Adam optimizer Kingma (2014). The initial learning rate for the network is set to $5 \times 10^{-4}$, while the swin-transformer Liu et al. (2021) backbone with layer normalization uses a learning rate scaled down by a factor of 0.03. Additional normalization was omitted due to the inherent design of the normalized transformer. The learning rate is scaled by a factor of 0.1 after epoch 2 and 5. For PascalVOC the batch size of image pairs is set to 8 and for Spair-71k the batch size is set to 5. Our validation set is obtained by taking 1000 image pairs per class. We use augmentations provided through the Albumentations package Buslaev et al. (2020). Specifically, we use Mixup Zhang et al. (2017), Cutmix Yun et al. (2019) and Random Erasing Zhong et al. (2020). The model is trained on one A100 GPU with 40 GB memory and takes about 9 hours for PascalVOC and 7 hours for SPair-71k. Other hyperparameters are listed in Table 1.

It is noteworthy that we only need 6 epochs to train, while BBGM Rolínek et al. (2020) needs 10, ASAR Ren et al. (2022) 16 and COMMON Lin et al. (2023) 16 epochs. Even though time per epoch might not be comparable, since the normalized transformer needs somewhat more time due to worse kernel fusion as compared to a vanilla transformer, this reveals significant possible training time savings on a more optimized normalized transformer implementation.

**Datasets.**  We train and test on PascalVOC and SPair-71k, the two most challenging current sparse keypoint matching datasets. We opted to forego e.g. Willow Object Class Cho et al. (2013) and other similar datasets since performance of previous works is already almost perfect there.

**PascalVOC:** We use PascalVOC Everingham et al. (2010) images with Berkeley annotations Bourdev & Malik (2009). Images are from 20 classes and are of size $256 \times 256$. Up to 23 keypoints are contained in each image. In order to be comparable to other works we use standard intersection filtering, i.e. when matching we only include keypoints that are in both images and discard outliers.

**SPair-71k:** The SPair-71k dataset Min et al. (2019) is a successor to PascalVOC and offers higher image quality and keypoint annotation and removal of problematic and poorly annotated image categories. It contains 70.958 image pairs. Images are taken from PascalVOC and Pascal3D+.

**Baselines.**  We compare our results with the highest-performing baselines from the literature, to the best of our knowledge.

**Results.**  Class-specific and overall results in terms of matching accuracy are presented in Table 2 for PascalVOC and in Table 3 for SPair-71k. Selected qualitative examples are shown in Figure 4.

On PascalVOC we outperform on average the baselines by 5.1%. We are better on 17 out of 20 image categories. On SPair-71k we overall outperform all baselines by 2.2% matching accuracy. We are better on 13 out of 18 image categories and second best on 3.

**Inference Speed**  We measure inference speed on a single NVIDIA GeForce RTX 4090 (batch size = 1), averaging over 1000 forward passes. The complete Normalized Matching Transformer processes each image pair in 44.4 ms, with the backbone + SplineCNN accounting for 39 ms and the two decoders for 5.4 ms.

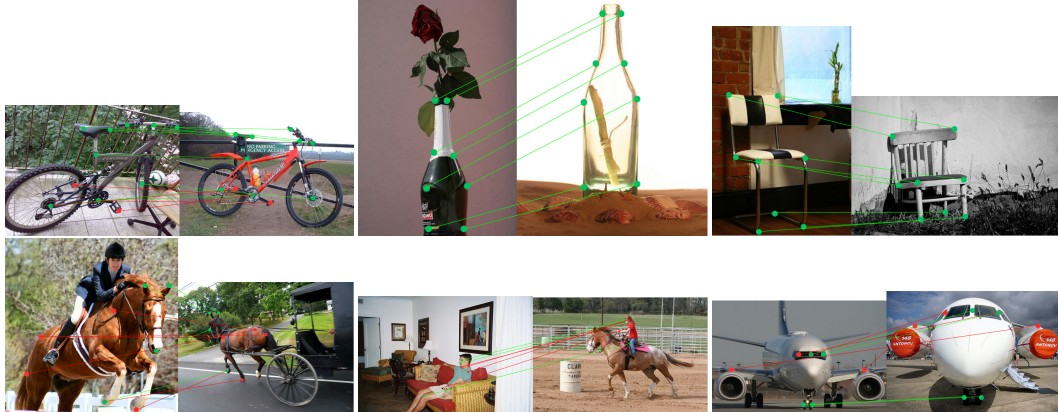

Figure 4: Qualitative results of selected keypoint matchings from the SPair-71k Min et al. (2019) dataset. The top row depicts perfect matchings, while the bottom row shows a few failure cases.

**Ablations.**  We provide ablations on our main architectural and training contributions on PascalVOC. The results are summarised in Table 4. We see that all contributions significantly add to final performance. The main performance driver are our losses, followed by improved backbone, normalized transformer and augmentations. Augmentations, while giving $1.2\%$, are still significant, but not overly so. The backbone ablation shows that we obtain better results on PascalVOC even when we employ the VGG backbone (i.e. 83.8%) also used by COMMON Lin et al. (2023), ASAR Ren et al. (2022) and BBGM Rolínek et al. (2020) and we even slightly outperform GMTR Guo et al. (2024) which uses a swin-transformer backbone.

We have also experimented with other simple pixel-wise augmentations like changing saturation value, random gamma, RGB shift etc., which however degraded performance.

Table 4: Ablation study on PascalVOC. We ablate augmentations, replacing the normalized transformer by a vanilla one, replacing InfoNCE and hyperspherical loss by cross entropy and replacing the swin-transformer backbone with VGG Simonyan & Zisserman (2014).

| Method | PascalVOC |
|---|---|
| NMT (FULL) | 88.7% |
| w/o augmentation | -1.2% |
| w/o layer loss | -0.8% |
| w/ vanilla transformer | -2.6% |
| w/ cross entropy Loss | -15.1% |
| w/ VGG16 | -4.9% |

## 5  CONCLUSION

We have introduced a hypersphere-centric paradigm for sparse keypoint matching, enforcing layer-wise normalization across all Transformer layers. Coupled with contrastive InfoNCE and hyperspherical uniformity losses, our model learns embeddings that align tightly across images while dispersing robustly within each image. Experiments on PascalVOC and SPair-71k show state-of-the-art accuracy and require $\geq 1.7\times$ fewer epochs to converge. These results underscore the impact of pervasive normalization and hyperspherical learning, with promising implications for future geometric and structured representation learning tasks.

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
