# OpenReview forum: "Normalized Matching Transformer"
_ICLR.cc/2026/Conference — ICLR 2026 Conference Withdrawn Submission_

### Official Review · Reviewer_9m5L · 2025-10-31

**Soundness:** 1
**Presentation:** 1
**Contribution:** 1
**Rating:** 2
**Confidence:** 4

**Summary:**

The paper proposes a novel model for keypoint matching between image pairs. The approach introduces unit hyperspherical normalization applied at each layer, combined with a global normalization step. Experimental results on two benchmark datasets demonstrate that the proposed method outperforms existing approaches.

**Strengths:**

* The proposed NTM model achieves better performance compared to related work on two datasets: Pascal VOC and SPair-71k.
* The normalization strategy has a positive impact, as evidenced by the ablation study.

**Weaknesses:**

* The authors described that the model is faster because it requires fewer training epochs. However, wall-clock time is a better metric for comparing efficiency between models.
* The Method section could be significantly improved. The current version does not clearly explain the motivation behind each model component, and the overall narrative looks like an assembly of separate modules rather than a cohesive design.
* Figure 2 presents several issues: the left portion appears blurred; it displays cosine similarity between the same feature vectors (f1 or f1), which should be zero since they are identical; and the font size is too small for readability.
* The proposed model shows a conceptual overlap with SuperGlue (Sarlin et al., 2020). Both methods employ an attentional graph neural network and a matching layer based on the assignment problem using the Sinkhorn algorithm. The paper does not sufficiently discuss the distinctions between the proposed method and SuperGlue, nor does it include a direct comparison in the experimental section.

**Questions:**

* What are the key differences between SuperGlue and the proposed approach?
* How does the model perform without the cross-attention mechanism or when using alternative visual feature extractors?
* In Equation (3), is there a typo? The summation should likely range from j = 1 to m, since the matrix C is of size m × m.

---

### Official Review · Reviewer_HReK · 2025-11-02

**Soundness:** 2
**Presentation:** 3
**Contribution:** 2
**Rating:** 2
**Confidence:** 4

**Summary:**

This paper proposes Normalized Matching Transformer (NMT) for sparse semantic keypoint matching between image pairs. The method combines a Swin transformer backbone, SplineCNN for geometric refinement, and a normalized transformer decoder with hyperspherical normalization at every layer. The training incorporates InfoNCE and layer-wise hyperspherical uniformity losses. The paper reports state-of-the-art results on PascalVOC (88.7%) and SPair-71k (86.7%), outperforming recent methods like GMTR and COMMON by 5.1% and 2.2% respectively, while achieving 1.7× faster convergence.

**Strengths:**

1. **Strong empirical results**: Substantial improvements on both benchmarks - +5.1% on PascalVOC, +2.2% on SPair-71k, with best performance on 17/20 and 13/18 categories respectively (Tables 2-3).
2. **Informative ablations**: Table 4 clearly shows loss function contributes most (-15.1%), followed by backbone (-4.9%) and normalized transformer (-2.6%), validating design choices.
3. **Faster convergence**: Only 6 epochs vs 10-16 for baselines, demonstrating improved optimization efficiency.
4. **Clear presentation**: Effective visualizations (Figures 1-3) and comprehensive implementation details (Table 1) aid reproducibility.

**Weaknesses:**

### Critical Issues

1. **Questionable problem relevance for ICLR 2026**: The paper exclusively evaluates on legacy sparse semantic keypoint matching benchmarks (PascalVOC from 2010, SPair-71k from 2019) without demonstrating practical applications or broader impact. The introduction and conclusion mention applications vaguely ("feature matching problem", "geometric tasks") but provide no concrete use cases, real-world deployments, or downstream task evaluations. For an ML-focused (not CV-focused) venue like ICLR, the paper should demonstrate why this problem matters in the field of AI in the present era and how the insights generalize beyond these specific benchmarks. Related dense matching methods (SuperGlue, LoFTR) have clear applications in SLAM, robotics, and AR/VR - what are the equivalent applications here?
2. **Undefined evaluation metric**: "Matching accuracy" is never formally defined. What exactly is "intersection filtering"? What percentage of keypoints are filtered?
3. **Unjustified scope restriction**: The method is restricted to sparse matching when core components (normalized transformer, contrastive losses) apply equally to dense matching. No justification provided for this limitation, no discussion of whether extension is possible, and no explanation of why sparse-specific methods are needed in the field of AI in the present era.
4. **Incremental novelty**: Combines existing techniques (Swin/SplineCNN/nGPT/InfoNCE/hyperspherical loss) with only minor addition of linear layer-wise weighting. No exploration of alternatives or theoretical justification for the simplest weighting scheme.
5. **Missing broader impact**: What insights transfer to other domains? Normalized features with cosine similarity are already standard in face recognition, contrastive learning, and retrieval. Contribution to broader ICLR community is unclear.

### Moderate Issues

1. **Training-inference mismatch**: Sinkhorn only at inference while training uses raw cosine similarities. Impact never analyzed.
2. **Incomplete details**: Missing Sinkhorn hyperparameters, temperature initialization, statistical significance (no error bars), and failure mode analysis despite showing failure cases (Figure 4).

**Questions:**

### Critical

1. **Can you demonstrate the method on at least one real-world application or downstream task?** (e.g., texture transfer, 3D shape correspondence, robotic manipulation)
2. **Can you extend to dense matching with results on HPatches or MegaDepth?** If not feasible, provide detailed technical discussion of what prevents extension.
3. **Please formally define "matching accuracy" with mathematical formula.** How many keypoints are filtered on average?
4. **What specific applications of sparse semantic matching cannot be addressed by dense matching or foundation models** (CLIP, SAM)?

### Secondary

1. Why not use differentiable Sinkhorn during training? What's the performance impact?
2. Your VGG16 result (83.8%) slightly outperforms GMTR with Swin (83.6%), suggesting loss is more important than backbone. Please clarify.

---

### Official Review · Reviewer_MqaQ · 2025-11-02

**Soundness:** 1
**Presentation:** 1
**Contribution:** 1
**Rating:** 2
**Confidence:** 3

**Summary:**

This paper proposes the Normalized Matching Transformer (NMT) for sparse keypoint matching. The method integrates a Swin Transformer backbone, a SplineCNN for geometric feature refinement, a normalized transformer decoder, and a combination of InfoNCE and hyperspherical losses. The authors report state-of-the-art results on PascalVOC and SPair-71k datasets, with notably faster convergence.

While the empirical results are strong, the paper suffers from fundamental issues regarding its core contribution, narrative, and experimental validation. The work is primarily presented as a composition of existing components from different domains (e.g., normalized transformers from NLP, SplineCNN from GNNs, hyperspherical losses), without delivering a novel insight or a clear, unifying principle for the problem of semantic keypoint matching. The narrative is largely experimental, reading like a technical report that documents a successful recipe but fails to provide a deeper understanding for the reader. Furthermore, the ablation studies are insufficient to substantiate the claimed contributions of key components beyond the use of a powerful backbone.

**Strengths:**

The reported performance on PascalVOC and SPair-71k is impressive and exceeds current state-of-the-art methods.

The training convergence speed is notably faster than several compared baselines, which is a practical advantage.

**Weaknesses:**

Lack of Conceptual Novelty and Insight: The main weakness of this paper is the questionable nature of its innovation. The architecture is a combination of well-known, off-the-shelf modules: a Swin backbone (from general vision), SplineCNN (from geometric deep learning), a normalized transformer (from recent NLP literature), and a standard contrastive loss (InfoNCE) paired with a hyperspherical uniformity loss. The paper positions the "pervasive normalization" as a key contribution, but this feels more like an engineering choice—applying a recently successful technique from another field—rather than an insight derived from an analysis of the semantic matching problem itself. The work does not answer why this particular combination is conceptually suited for keypoint matching beyond the fact that it yields higher numbers.

Inadequate Narrative and Scholarly Presentation: The paper is written as a sequence of technical decisions and experimental results. It fails to build a compelling scientific narrative. It does not sufficiently motivate why this specific combination of components is necessary from a theoretical or intuitive perspective, nor does it critically discuss the limitations or failure modes of the proposed approach. A reader is left with a "what" (the results) but not a clear "why" (the underlying reason for its success), which limits the paper's value to the community as more than a data point for a specific configuration.

Insufficient and Unconvincing Ablation Studies: The ablation study in Table 4 is critically flawed and does not adequately support the authors' claims.

It does not include an ablation for the InfoNCE loss, which is claimed to be a central part of the method. The ablation only replaces the combined (InfoNCE + HS) loss with a cross-entropy loss, which is a drastic change. The individual contribution of InfoNCE versus the hyperspherical loss remains completely unquantified.

The results strongly suggest that the Swin-Large backbone is the primary driver of performance. The ablation shows that using a VGG backbone causes a -4.9% drop, while all other modifications (removing augmentation, layer loss, or using a vanilla transformer) result in smaller deficits (≤ -2.6%). This raises a serious question: how much performance gain is truly attributable to the novel aspects of the matching architecture (SplineCNN, normalized transformer, losses) versus simply using a much more powerful feature extractor? The current experiments cannot rule out the possibility that the marginal gains from other components are a result of cherry-picking or hyperparameter tuning rather than a fundamental improvement.

**Questions:**

What is the specific, isolated performance contribution of the InfoNCE loss, separate from the hyperspherical loss?

Given the massive performance gap introduced by the backbone switch (Swin vs. VGG), can you provide a more detailed ablation that clearly disentangles the performance contributions of the SplineCNN, the normalized transformer, and the loss functions when using the same backbone?

Beyond achieving high scores, what specific challenge in semantic keypoint matching does the "hypersphere-centric paradigm" solve that previous methods struggled with? Can you provide a qualitative or quantitative analysis that demonstrates this?

---

### Official Review · Reviewer_Ygxx · 2025-11-03

**Soundness:** 2
**Presentation:** 2
**Contribution:** 1
**Rating:** 2
**Confidence:** 4

**Summary:**

This paper proposes Normalized Matching Transformer for sparse matching between image pairs.
The architecture consists of a feature backbone, Spline CNN for feature refinement, and a normalized transformer to yield the matching features.
The authors propose the hyperspherical normalization strategy as the central component of NMT, enforcing unit-norm embeddings at every transformer layer, and yielding more discriminative keypoint features by a combined contrastive InfoNCE loss.
Quantitative results on the PASCAL-VOC and SPair-71k datasets show that NMT obtains SoTA results, with the use of InfoNCE and hyperspherical loss showing the highest performance gain in the ablation experiments.

**Strengths:**

- The overall architecture is simple and straightforward - feature extractor, feature refinement, and feature matching.

- The proposed loss function (InfoNCE + hyperspherical loss) is effective and leads to strong performance gains.

**Weaknesses:**

- Weak algorithmic novelty. The paper builds on well-built foundations of strong feature extractor, feature refinement via Spline CNN, and transformer-based feature matching via alternating self- and cross- attention of features. While the InfoNCE loss and hyperspherical loss brings about dramatic improvements, the introduction of such contrastive losses is also a well-known concept in image matching [1].

- Lack of comparative experiments across a large body of baseline work on semantic matching. The current evaluation setting only considers the case when all the source and target keypoints are known, and thus the corresponding metric (accuracy) is being used. However, a larger body of work, e.g., [2][3][4][5], perform semantic matching where only the source keypoints are known, and the given architecture of NMT can be easily applied to such settings as well (e.g., by performing top-1 similarity for each source keypoint).

- Lack of application on related sparse geometric matching work. SuperGlue is in fact a sparse matching method, as they rely on keypoints  selected from SuperPoint. In that case, how would the proposed method fare in sparse matching scenarios given geometric matching datasets such as HPatches, Aachen Day-Night or MegaDepth datasets? The authors refer to SuperGlue and LightGlue as dense keypoint matching methods, but LoFTR is the only dense keypoint matching method among the three methods listed in L191. If the authors meant to use 'sparse' to mean 'a little number of keypoints', that would have to be specified more clearly.

- Wrong writing formatting in the citations. All citations seem to be in `\cite{}` instead of `\citep{}` or `\citet{}`.

[1] Choy et al., "Universal Correspondence Network", 2016 \
[2] Cho et al., "Cost Aggregation Transformers for Visual Correspondence", 2021 \
[3] Kim et al., "TransforMatcher: Match-to-Match Attention for Semantic Correspondence", 2022 \
[4] Zhang et al., "A Tale of Two Features: Stable Diffusion Complements DINO for Zero-Shot Semantic Correspondence", 2023

**Questions:**

- What would the authors propose as the main algorithmic novelty of NMT?

- How does NMT fare against semantic matching baseline methods, which are also evaluated on SPair-71k and PASCAL-VOC datasets?

- How does NMT perform compared to other sparse geometric-matching methods?

---

### Note · Authors · 2026-05-19

I have read and agree with the venue's withdrawal policy on behalf of myself and my co-authors.

---

### Meta-Review · Area_Chair_YHKc · 2026-01-07

**Summary:**

Ygxx:

* Weak algorithmic novelty. Loss functions used by authors are similar to ones previously known in image matching.

* Lack of comparative experiments across a large body of baseline work on semantic matching.

* Has asked for clarity about the contribution and more experiments. "


MqaQ:

* Lack of Conceptual Novelty and Insight:
* draft is missing ""why""
* ablation needs to improve
* questioned if improvement is due to stronger backbone"

HReK

Questionable problem relevance for ICLR

9m5L

Unclear about difference of contribution (superGlue vs draft)

**Reviewer Concerns:**

None.

**Reviewer Scores:**

Very less chance of any change.

Reviewer Ygxx has raised concerns on both novelty and experimental verification, and assigned rank-2. There is very little chance that this score would be revised to accept.

Reviewer MqaQ also raised questions about novelty and insufficient ablations, but also was critical of the writing. Change of score would have required major improvement in draft.

HReK questioned whether problem has relevance in ICLR 2026, and also was concerned about having incremental novelty.

9m5L also said that narrative needs to improve and was of view that there is conceptual overlap with SuperGlue.

---

### Decision · Program_Chairs · 2026-01-26

Reject